# GluN2A and GluN2B N-Methyl-D-Aspartate Receptor (NMDARs) Subunits: Their Roles and Therapeutic Antagonists in Neurological Diseases

**DOI:** 10.3390/ph16111535

**Published:** 2023-10-30

**Authors:** Amany Digal Ladagu, Funmilayo Eniola Olopade, Adeboye Adejare, James Olukayode Olopade

**Affiliations:** 1Department of Veterinary Anatomy, University of Ibadan, Ibadan 200284, Nigeria; amanykladagu@ymail.com (A.D.L.); jo.olopade@mail1.ui.edu.ng (J.O.O.); 2Developmental Neurobiology Laboratory, Department of Anatomy, College of Medicine, University of Ibadan, Ibadan 200284, Nigeria; 3Department of Pharmaceutical Sciences, Philadelphia College of Pharmacy, Saint Joseph’s University, Philadelphia, PA 19131, USA

**Keywords:** GluN2A and GluN2B, N-methyl-D-aspartate receptors (NMDARs), antagonists, compounds, neuroprotection, neurodegeneration

## Abstract

N-methyl-D-aspartate receptors (NMDARs) are ion channels that respond to the neurotransmitter glutamate, playing a crucial role in the permeability of calcium ions and excitatory neurotransmission in the central nervous system (CNS). Composed of various subunits, NMDARs are predominantly formed by two obligatory GluN1 subunits (with eight splice variants) along with regulatory subunits GluN2 (GluN2A-2D) and GluN3 (GluN3A-B). They are widely distributed throughout the CNS and are involved in essential functions such as synaptic transmission, learning, memory, plasticity, and excitotoxicity. The presence of GluN2A and GluN2B subunits is particularly important for cognitive processes and has been strongly implicated in neurodegenerative diseases like Parkinson’s disease and Alzheimer’s disease. Understanding the roles of GluN2A and GluN2B NMDARs in neuropathologies provides valuable insights into the underlying causes and complexities of major nervous system disorders. This knowledge is vital for the development of selective antagonists targeting GluN2A and GluN2B subunits using pharmacological and molecular methods. Such antagonists represent a promising class of NMDA receptor inhibitors that have the potential to be developed into neuroprotective drugs with optimal therapeutic profiles.

## 1. Introduction

The synthesis and assembly of N-methyl-D-aspartate receptor (NMDAR) subunits occur in the rough endoplasmic reticulum (RER), after which they are conveyed to the dendrites by vesicles where insertion into the spines takes place [1]. They are proportioned into three different pools, based on their activities: i. synaptic pool; ii. extrasynaptic pool; and iii. non-synaptic pool (cell bodies and dendrites containing NMDARs) [2]. The location of the receptor, either synaptic or extrasynaptic, explains the dual nature of NMDARs in function and pathology. Synaptic NMDARs are known to play a major role in the initiation of survival signaling, whereas extrasynaptic NMDARs are related to deregulation of Ca^2+^ and cell death [3]. NMDA receptors’ surface expressions, coupled with their subunit compositions, are dynamically controlled by the activities of hippocampal neurons [4] and other associated neurons in the cortex. Depending on the subunit composition, NMDARs are trafficked, placed, or displaced from the synaptic sites through various processes [5]. NMDAR (Figure 1) is essential for synaptic plasticity and neuronal development. They influence the migration of neurons to their final destinations in the developing brain [6]. They are involved in guiding neurons along radial glial fibers and helping them reach their appropriate layers within the cerebral cortex [7]. It is therefore not unexpected that aberrant trafficking and mis-localization of the subunits have been reported in the vast majority of brain disorders and pathological conditions [8]. Aberrant NMDA receptor activity and subunit composition explains why neurotransmission patterns are impaired in Alzheimer’s disease and Parkinson’s disease (PD) and other neurodegenerative diseases because NMDARs in the hippocampus are essential for memory acquisition and memory consolidation [9,10]. Evidently, NMDA receptors are involved in the development and maintenance of dendritic spines [10]. Proper spine formation is essential for the structural and functional organization of neural circuits. 

### 1.1. NMDARs’ Subunit Composition and Subtype Selectivity

Recent findings have also revealed the complexity of NMDA receptor subunit composition. Besides the well-known GluN1 and GluN2 subunits, there is growing recognition of the importance of GluN3 subunits and their role in NMDA receptor function. Understanding the subunit composition and diversity of NMDA receptors has implications for drug development and the targeting of specific receptor subtypes [12]. NMDARs are made up of two obligate GluN1 subunits, ubiquitously expressed from embryonic stage E14 to adulthood in rats (Figure 2); because of the widespread CNS distribution of NMDARs [13,14], they are encoded by a single gene which consists of eight distinct isoforms due to alternative splicing, GluN1-1a–GluN1-4a and GluN1-1b–GluN1-4b. Both GluN1-a and GluN1-b isoforms possess overlapping expression patterns with relative abundance that vary in different anatomical regions. The variations in their development and regions depend on the type of GluN1 isoform [15,16]. Some isoforms are notably expressed in the hippocampus, at high amounts in the principal cell layers (CA1-3 layers, dentate gyrus, and granule cells), while some are extensively distributed throughout the brain, including the thalamus and the cerebellum. GluN1-a isoforms in the hippocampus are found in all principal cells, while GluN1-b isoforms are mostly confined to the CA layer [17].

The GluN2 subunit, of which there are four types (2A–D) (Figure 2), has patterns of expression that vary prominently in both time and location, that is, during development and in different regions. These are illustrated rats (Figure 2). The primary types of NMDARs expressed at synapses at early developmental stages are GluN1/2B diheteromers, as shown by high levels of sensitivity of excitatory postsynaptic currents (EPSCs) to selective GluN2B inhibitors CP101, 606, and ifenprodil [18]. GluN2A progressively increases after birth in the brain until adulthood (predominantly in the forebrain) (Figure 2), though its expression is abundant in the entire CNS. While GluN2B is expressed at an early stage and widely distributed, its expression peaks around postnatal day seven [7] (Figure 2) [19], a stage at which there is a sharp rise in GluN2A expression, but GluN2B gradually becomes confined to areas of the forebrain where it remains at high levels (Figure 2). This phenomenon is known as ‘the developmental GluN2B-GluN2A switch’, whereby GluN2A expression progressively rises while expression of GluN2B remains constant. The molecular and cellular bases are not fully understood but seem driven by activities, though there are some occasions where GluN2B’s expression remains in the prefrontal cortex during adolescence at very high levels [20]. This ‘shift phenomenon’ in NMDARs’ subunit composition occurs during forebrain development from almost exclusively GluN2B-NMDARs to populations of GluN2A-NMDAR representations, indicating both diheteromeric GluN1-GluN2A receptors and triheteromeric GluN1-GluN2A-GluN2B receptors [21].

The complexity of GluN2A and GluN2B ratio distributions in different sub-regions of the brain (Figure 2) can explain their unique pathological and physiological effects [22,23]. GluN2A and GluN2B ratio variations in individuals with healthy brains may also explain their different LTP/LTD capacities and performance in long-term memories. GluN2A and GluN2B ratio adjustment is generally considered to participate in increasing capacity for synaptic plasticity as well as synaptic maturation. This adjustment strategy of GluN2A and GluN2B could be put into consideration in treating patients that carry heterozygous GluN2A or GluN2B mutations that have related mental retardation or epilepsy [24]. Abnormalities in NMDA receptor function or mutations in NMDA receptor genes have been implicated in neurodevelopmental disorders such as autism spectrum disorders and intellectual disabilities [25]. Understanding the roles of NMDA receptors in development is crucial for elucidating the underlying mechanisms of these conditions.

The GluN2 subtype regulates a wide variety of NMDAR biophysical properties; this includes open probability, deactivation kinetics, and agonist affinity, which in turn influence downstream signaling and synaptic NMDAR-evoked currents [26]. GluN2A and GluN2B subunits have a higher degree of single-channel conductance compared with Glu1/2C and GluN1/2D subunits [27]. An increase in GluN2A/GluN2B ratio impedes retrieval-dependent memory destabilization of a weak memory trace and consequently thwarts its modification, which is similar to re-consolidation-resistant fear memories. They are critical for LTP induction levels, though there are obvious concerns that an increase in receptors containing GluN2B relates to excitotoxicity [28]; increasing GluN2A could be less excitotoxic. In addition, some specific cognitive functions may depend on receptors containing both GluN2A and GluN2B because they play special physiological functions in several processes of the CNS [28]. Recently, the relative protein levels of NMDAR subunits in crude were assessed, and microsomal fractions derived from the cerebral cortex, hippocampus, and cerebellum of adult mice were examined. Interestingly, a substantial presence of GluN2D protein in adult brains was observed, even though its transcription levels decrease following the early postnatal stages [12]. NMDA receptors are active during critical periods of brain development [29]. These are specific time windows during which certain brain functions, such as vision or language, are especially sensitive to environmental input. Disruptions in NMDA receptor function during critical periods can have long-lasting effects on brain development and function.

### 1.2. GluN1/2A and GluN1/2B Diheteromer and Triheteromer Receptors

Strong evidence has shown that the coexistence of diheteromers and triheteteromers within a single cell or at a single synapse contributes to the functional diversity of the postsynaptic response [30]. Therefore, the mechanism of action and subunit selectivity are of paramount importance, when considering the application of a pharmacological ligand in neurophysiological studies because several central synapses in the adult hippocampus and cortex are made up of mixed populations of GluN1/2A, GluN1/2B, and triheteromeric GluN1/2A/2B receptors [31,32]. Multiple studies have also revealed that the functional and pharmacological features and characteristics of triheteromeric GluN1/2A/2B receptors are very different from those of diheteromeric GluN1/2A and GluN1/2B receptors. The developmental change that takes place from the diheteromeric GluN1/2B form of receptors to triheteromeric GluN1/2A/2B receptors [33] supports synaptic plasticity alterations and neuronal circuit maturation [34]. In addition, heterogeneity among NMDA receptor subunits and assemblage of various receptor subtypes that possess definite functional characteristics allow accurate synaptic response tuning and permit variations in physiological roles and functions of the receptors during neuronal development [35].

Dihetereromeric GluN2A-NMDARs (GluN1/2A receptors) are highly sensitive to Mg^2+^ block and Ca^2+^ permeability. Thus, they have a higher probability of channel openings with a faster decay rate than diheteromeric GluN2B-NMDARs (GluN1/2B receptors) [36]. Triheteromeric receptors are expressed throughout the brain in neurons but are at an abundant density within the cells of the hippocampus, hypothalamus, and amygdala [30]. Subsequently, utilizing the distinctive features of triheteromeric GluN1/GluN2A/GluN2B channels will help in comprehending physiological synaptic events and changes in synaptic activities during neuropathologies [37]. Previous studies have suggested that approximately two-thirds of NMDARs’ population in the hippocampus of rodents may be triheteromeric [38]. Several investigations have shown that a great amount of naïve triheteromeric NMDA receptors are assembled from two GluN1subunits and two different GluN2 subunits. Specifically, these studies have shown that triheteromeric GluN1/GluN2A/GluN2B populations are responsible for over 50% of the entire NMDA receptors in the hippocampus and cortex of the adult rodent brain, thus revealing their peculiar gating and pharmacological hallmarks [39]. A distinctive intra- and inter-subunit interface also exists in GluN1/2A/2B NMDARs triheteromers, which are absent in GluN1/2A and GluN1/2B NMDARs diheteromers [40]. Therefore, there are reasons for targeting either of the subunits, GluN2A or GluN2B, though more studies need to be conducted to discern if the advantages of targeting one exceeds the other. A promising strategy could be in designing compounds selective for triheteromeric GluN1/GluN2A/GluN2B receptors. However, selectively targeting this receptor subtype without interference from others could be quite difficult [41]. GluN1/2A/2B-NMDARs’ distinct structure and function generate opportunities for the design and development of novel antagonists selective for triheteromeric over diheteromeric receptors. Such selectivity can provide novel and strategic ways of treating neurological and psychiatric disorders [42].

### 1.3. Synaptic and Extrasynaptic GluN2 NMDA Receptors

NMDA receptors are central to synaptic plasticity, which is the ability of synapses to strengthen or weaken over time in response to neuronal activity. During development, synaptic plasticity allows for the refinement of neural circuits and the adaptation of the brain to sensory inputs. NMDA receptor-dependent long-term potentiation (LTP) and long-term depression (LTD) are crucial mechanisms underlying synaptic plasticity and learning [42,43]. NMDA receptors also play a role in experience-dependent plasticity, where the neural circuits are shaped by sensory experiences [44]. During critical periods of development, sensory inputs (e.g., visual, auditory) are required to fine-tune and maintain synaptic connections. NMDA receptors are involved in these processes, allowing the brain to adapt to the environment.

Two models describe the differential contributions of NMDARs to neuronal survival and excitotoxicity. The “localization model”, which postulates that activation of synaptic NMDARs is neuroprotective whereas activation of extrasynaptic NMDARs is neurotoxic [45,46], and the “subunit composition model”, which proposes that the subunit make-up of NMDARs determine if their activation has neuroprotective or neurotoxic consequences. Receptors found at the pre-synaptic locations contribute to neuroplasticity and synaptic transmission, while receptors at postsynaptic locations contribute only to the regulation of plasticity [47]. Both pre-synaptic and post-synaptic receptors are involved in activation of neuronal survival and protection genes. Higher concentrations of glutamate are necessary for the activation of extrasynaptic receptors, which are located on dendrites. These receptors support the GluN2B subunit, which plays a role in neurotoxicity and regulation of neuronal cell death [48]. NMDARs containing NR2B subunit are enriched in the extrasynaptic receptor population, which makes them potential candidates for supporting disease-associated ‘background’ NMDAR activation [49]. Mitochondrial membrane potential loss and cell death are observed due to stimulation of extrasynaptic NMDA receptors, whereas synaptic NMDA receptors have been reported to be neuroprotective. The influx of calcium through extrasynaptic NMDARs may cause mitochondrial dysfunction, which eventually leads to neuronal death. Therefore, neurotoxicity is not only mediated by Ca^2+^ overload but also by Ca^2+^ influx through extrasynaptic receptors which are mainly deleterious to neurons. Synaptic NMDA receptor activation has anti-apoptotic activity, whereas extrasynaptic NMDA receptor activation elicits mitochondrial membrane potential loss, a progenitor for glutamate-induced toxicity. Therefore, antagonism of specific extrasynaptic NR1/NR2B receptors in neuropathological conditions may have effective neuroprotective properties, without interfering with normal synaptic NR1/NR2A-mediated events [50]. A promising therapeutic strategy is to selectively target extrasynaptic NMDAR signaling which are characterized by GluN2B subunits, while sparing synaptic signaling [51].

The presence of NMDARs containing GluN2A and GluN2B subunits at the synapse is balanced by their associations with the postsynaptic density (PSD) subfamily of membrane-associated guanylate kinases (MAGUKs). MAGUKs are typical neuronal scaffolding proteins that participate in multiple signaling cascades by binding to numerous interacting proteins [52]. These include PSD-93, PSD-95, SAP102, and SAP97 [30,53]. Synaptic and extrasynaptic localization of GluN2A and GluN2B is determined by PSD-95 and SAP-102; the heightened expression of PSD-95 in the developmental period probably competes with SAP102 for insertion into the postsynaptic density and displaces SAP-102 and GluN2B outside of synapses [54,55]. GluN2A’s mobility is less than GluN2B’s because of its tendency to interact with PSD-95, and it is confined to synapses in adults [56]. GluN2A and GluN2B’s different subcellular locations, which are induced by PSD-95 and SAP-102, are probably responsible for their differences in signaling.

GluN2A has higher glutamate affinity than GluN2B, and release of synaptic glutamate in adult neurons leads to higher stimulation of NMDARs containing GluN2A than GluN2B. These seemingly contradictory properties are most likely the result of the segregated subcellular localization of these subunits, which favors activation of GluN2A (expressed predominantly at synaptic sites) over GluN2B (enriched predominantly at extrasynaptic sites). This model solely depends on data from the use of selective NMDARs containing GluN2B inhibitors like ifenprodil, CP101, 606, or Ro25-6981. Thus, only about 30% of the NMDAR-mediated excitatory postsynaptic current (NMDAR-EPSC) can be blocked by ifenprodil (Figure 3) in adulthood, showing the presence of GluN2A which are predominant at synaptic sites [48].

Consistent with the importance of precise NMDAR regulation for proper synaptic and neuronal functions, several neuronal disorders are characterized by altered NMDAR subunit expression and synaptic or extrasynaptic localization [48]. Cognitive decline in neuronal aging and neurodegenerative diseases such as AD may be the result of an imbalance between synaptic and extrasynaptic NMDA receptors [57]. Pathological conditions can increase the activation and expression of extrasynaptic NMDARs and therefore favor pro-death pathways. Extrasynaptic NMDA receptors may be stimulated by spillover of glutamate from synapses or from ectopic release of glutamate, a negative influence on the neurons, following over-activation of extrasynaptic NMDA receptors, which can eventually lead to damage and death of cells; this is evident in some major diseases of the nervous system [58]. In fact, a vast majority of pathological conditions and neurodegenerative diseases, such as Alzheimer’s, Parkinson’s, and Huntington’s, may involve an abnormal increase in the number of extrasynaptic NMDARs and/or signaling of same [46]. This rising knowledge of the roles of extrasynaptic NMDARs in pathologies offers researchers hope in comprehending the etiology of major diseases of the nervous system [54].

Studies have also shown that a possible relationship exists between a decrease in protein expression of NMDA in regions such as the hippocampus during senescence and impaired memory function [59]. This decrease involves a reduction in the expression of GluN2A and GluN2B in the hippocampus [60,61]. This happens with a change in the localization of GluN2B from the synapse to extrasynaptic sites [62]. GluN2B NMDA receptors are particularly mobile and translocate outside of synapses to extrasynaptic sites, a process which may increase with aging [57]. Since the GluN2B subunit is present [9] in extrasynaptic NMDA receptors and Tau overexpression has also been implicated in the overactivation of extrasynaptic NMDA receptors [63], it could be considered a potential target for the treatment of neurodegenerative disorders related to aging such as AD. Reduced glutamate uptake has also been associated with extrasynaptic NMDA receptors at the hippocampal CA1 synapse of aged rats [62]. Kumar et al. [64] have proposed a possible relationship between a decrease in NMDARs during senescence and impairment of memory function. Several other studies have also shown a decrease in expressions of NMDA receptor proteins during senescence in areas such as the hippocampus [57]. NMDA receptors are also involved in the regulation of neural stem cell proliferation and differentiation [65]. During early brain development, these receptors help control the balance between self-renewal and differentiation of neural stem cells. Activation of NMDA receptors influences the fate of neural precursor cells, guiding them to become specific types of neurons or glial cells Additionally, NMDA receptors are crucial for the formation of synapses, which are the connections between neurons [66]. As neurons extend their axons and dendrites, they form synapses with other neurons. The activation of NMDA receptors at these developing synapses is essential for the establishment of functional connections. This process is fundamental for the development of neural circuits and the wiring of the brain [67].

### 1.4. The Ideal Clinically Tolerated NMDA Receptor Antagonist

An NMDAR antagonist that is clinically tolerated would not cause drowsiness, hallucination, or coma in patients, but would spare normal neurotransmission while blocking the havoc of the excessive activation of NMDA receptors [68,69]. The ideal antagonist would be a compound that exhibits inhibitory potencies in the low micromolar range, allowing the physiological function of the receptors, and at the same time inhibiting them upon overactivation or upregulation [70]. The biophysical properties of the compound would enable it to differentiate precisely between low-level chronic pathological activation of NMDA receptors in Alzheimer’s disease (‘‘noise’’) and physiological synaptic activation of NMDA receptors (‘‘signal’’) [71,72]. At therapeutically relevant concentrations, the compound should have the ability to suppress the pathological background ‘‘noise’’ and at the same time preserve and enhance the normal physiological synaptic NMDA receptor-mediated plasticity ‘‘signal’’ [73]. NMDAR antagonists may be able to enhance cognition effectively by selective inhibition of the ‘pathological activation’ while preserving NMDA receptors’ normal ‘physiological activation’ [15]. Some drugs do not match the required properties and therefore eventually fail clinical trials because of unacceptable side effects (hallucinations, drowsiness, and sometimes coma); they display competition with either glutamate or glycine at the agonist binding sites which block normal functions [74,75,76]. Such drugs substantially interfere with normal synaptic transmission [76].

### 1.5. NMDA Receptor Antagonists’ Recent Advances

NMDA (N-methyl-D-aspartate) receptor antagonists have been the subject of extensive research and development due to their therapeutic potential in various neurological and psychiatric disorders. Here are some reports and recent advances related to NMDA receptor antagonists:1.Ketamine and Esketamine for DepressionNMDA receptors play crucial roles in brain development, and recent research has shed light on their functions in processes like synapse maturation, dendritic spine formation, and neuronal migration. Dysregulation of NMDA receptor signaling during development has been linked to neurodevelopmental disorders [77]. Innovative therapies targeting NMDA receptors are in development for conditions like stroke, traumatic brain injury, and neurodegenerative disorders. These therapies aim to modulate NMDA receptor activity to promote neuroprotection and functional recovery [78] NMDA receptor-based therapies are being explored for a range of disorders. For example, ketamine, an NMDA receptor antagonist, has shown promise in the treatment of treatment-resistant depression. However, the precise mechanisms of action and long-term effects of these treatments are still areas of active research [79]. Scientists are actively investigating compounds that can modulate NMDA receptor function. This includes developing allosteric modulators that can fine-tune NMDA receptor activity. Such modulators may have therapeutic potential in various neurological conditions [80].Esketamine, a derivative of ketamine, has also been developed and approved for use in depression treatment [81]. Research has focused on optimizing dosing regimens, administration routes, and long-term safety profiles of ketamine and esketamine. Efforts are underway to understand the mechanisms underlying their antidepressant effects and to expand their use in clinical practice [82].2.Memantine in Alzheimer’s DiseaseMemantine, an NMDA receptor antagonist, has been approved for the treatment of Alzheimer’s disease. It helps regulate glutamate signaling and mitigate excitotoxicity in neurodegenerative conditions [83]. Ongoing research explores the potential of memantine as a disease-modifying agent in Alzheimer’s disease and investigates its use in combination with other therapies. Researchers are also working on developing more selective NMDA receptor modulators with improved safety profiles. Memantine exhibits voltage dependency, uncompetitive antagonism, preferential inhibition of extrasynaptic receptors, partial trapping, affinity for the PCP-binding site, and mode of fast off-rate (a property that is intrinsic to the drug-receptor complex, not affected by the concentration of the drug). A major contributor to a drug’s low affinity for the channel pores is a relative fast off-rate [84]. Memantine is both neuroprotective and offers symptomatic improvement by the same MOA, i.e., moderate-affinity NMDA receptor channel blockade [70].3.NMDA Receptor Modulators in SchizophreniaDysregulation of NMDA receptors has been implicated in schizophrenia. Several compounds that modulate NMDA receptor function are being investigated as potential treatments for the disorder [85]. Recent studies have highlighted the potential of glycine-site modulators and other NMDA receptor-targeting agents in managing schizophrenia symptoms. Research continues to refine these compounds and explore their clinical efficacy. Additionally, advancements in structural biology techniques have allowed researchers to obtain high-resolution structures of NMDA receptors. These structures provide valuable insights into the receptor’s function and have the potential to inform drug design efforts [86,87]. These studies have deepened our understanding of how dysfunction in NMDA receptors contributes to neurological disorders. Notably, NMDA receptor hypofunction has been implicated in schizophrenia, and some studies have explored NMDA receptor-based treatments for the condition [88].4.NMDA Receptor Antagonists in Pain ManagementNMDA receptor antagonists are being explored as treatments for chronic pain conditions, including neuropathic pain and fibromyalgia. These drugs aim to reduce pain perception by blocking NMDA receptor activity [89] Researchers are investigating novel NMDA receptor antagonists with improved pharmacokinetics and safety profiles. Combining these drugs with other analgesics in a multimodal approach is also a focus of recent studies.5.NMDA Receptor Antagonists in NeuroprotectionNMDA receptor-based therapies are being studied for their neuroprotective effects in conditions such as stroke, traumatic brain injury, and neurodegenerative diseases. Modulating NMDA receptor activity can help mitigate excitotoxicity and neuronal damage. Recent studies have shown that ergotamine demonstrated a potent inhibitory effect specifically on the NR1a/NR2A subunit within the various subtypes of NMDAR. This inhibitory action serves to effectively curtail the excessive influx of calcium ions, thereby acting as a protective measure against neuronal cell death [90]. Advances in understanding the role of NMDA receptors in neuroprotection have led to the development of potential therapeutic interventions. Researchers are exploring the use of NMDA receptor antagonists as part of comprehensive neuroprotective strategies.6.Subunit-Selective NMDA Receptor ModulatorsRecent research has focused on developing subunit-selective NMDA receptor modulators. These compounds target specific types of NMDA receptors, such as GluN2A or GluN2B-containing receptors, for more precise therapeutic effects [91]. Subunit-selective modulators are being investigated for their potential in treating various neuropsychiatric disorders while minimizing side effects associated with non-selective NMDA receptor antagonists.7.Combination TherapiesCombining NMDA receptor-related therapies with other pharmacological agents is an emerging approach in clinical practice. These combinations aim to enhance therapeutic outcomes while minimizing side effects [92]. Ongoing research evaluates the safety and efficacy of combination therapies involving NMDA receptor modulators, especially in conditions like depression and chronic pain.

These recent advances reflect the evolving landscape of NMDA receptor antagonist research and its potential impact on the treatment of a wide range of neurological and psychiatric disorders. Continued investigation and clinical trials are essential to refining these therapies further and expanding their clinical applications.

### 1.6. GluN2A and GluN2B NMDA Receptor Antagonists

Channel blockers of NMDA receptors have shown robust neuroprotective outcomes in animal models of CNS disorders that exhibit excessive activation of NMDA receptors such as epilepsy, traumatic brain injury (TBI), and stroke [46]. The GluN2 subunits have received substantial interest as potential targets for therapy since they control the physiological functions of NMDA receptor subtypes [41]. Studies have proposed that NMDA receptors’ stimulation leads to a decrease in neurogenesis, while antagonism of NMDA receptors reverses this effect.

Selective antagonism of NMDA receptors containing GluN2B subunits has proven to be effective in curbing the pathological roles of excitotoxicity in the progression of various disorders such as AD, PD, neuropathic pain, and depression [93,94]. Ifenprodil, an antagonist of GluN2B containing NMDARs, was the first subunit-selective NMDA receptor antagonist to be discovered. It represents a class of non-competitive, voltage- and use-dependent inhibitors of NMDA receptors [73]. Inhibition of GluN1/GluN2B receptors by ifenprodil has a high affinity of about 200 to 400-fold, compared to different combinations of GluN1/GluN2 subunits [95]. Crystallographic studies have shown that ifenprodil binds precisely at the interface between amino-terminal domains (ATDs) of GluN1/GluN2 heterodimers [96]. Raybuck et al. (2017) [97] reported that ifenprodil reversed functional loss by rescuing spine loss in HIV-associated neurocognitive disorder (HAND). Increased spine loss in the hippocampus and medial prefrontal cortex correlates with impairment in learning [98]. Indeed, therapies that increase dendritic spines of the hippocampus can facilitate LTP-induction and enhancement of hippocampal-dependent learning and memory [99]. Felbamate has been observed to exhibit a specific preference for targeting the NR2B subunit of the NMDA receptor. NR2B mutations have been suggested as contributing to the development of epilepsy in mouse models. Felbamate is widely recognized for its effectiveness as an additional therapy in the treatment of challenging conditions such as intractable partial seizures. However, it is important to note that the use of felbamate as an adjunctive therapy in partial epilepsy has been constrained due to the occurrence of severe adverse effects, including aplastic anemia and liver disease [100]. Neu2000, alternatively known as nelonemdaz or salfaprodil, is a compound derived from sulfasalazine and belongs to the class of negative allosteric modulators targeting GluN2B-containing NMDARs and has been specifically designed to mitigate NMDAR-mediated excitotoxicity as well as free radical toxicity [101].

NP10679, developed by NeurOp Inc. in Atlanta, GA, USA, is designed to counteract minor changes in extracellular pH that happen during conditions such as strokes and traumatic brain injuries. It is noteworthy that extracellular protons can strongly inhibit NMDARs, impacting the activity of NMDARs containing the GluN2B subunit [102,103]. Nerinetide, alternatively referred to as NA-1 or Tat-NR2B9c, is a peptide composed of the Tat sequence followed by the last nine C-terminal residues of the NMDAR-GluN2B subunit, which contains the PDZ ligand. This compound is currently under development for use in treating ischemic stroke, traumatic brain injury (TBI), and subarachnoid hemorrhage. Importantly, it has been shown to be both safe and efficacious in the treatment of patients who experience iatrogenic stroke following endovascular aneurysm repair [104]. ZL006 has been documented as being capable of disassembling the GluN2B-PSD95-nNOS complex, and it has also been observed to penetrate the blood–brain barrier without impacting the regular functioning of NMDARs and nNOS [105].

Amidfar et al. (2019) [93] have shown that GluN2B-containing NMDARs blockade by Ro 25-6981 causes inhibition of nNOS activity, resulting in stimulation of adult neurogenesis in the hippocampal region of the brain [106,107]. It also enhances retrieval of spatial memory in rats. In addition, it facilitated formation of spatial memory in mice [108]. There is therefore an urgent need for identification of GluN2B antagonists that are potent, highly selective, and with good oral bioavailability since NMDA receptor subunits containing GluN2B have proven to be a target of therapy for neurodegenerative diseases [109,110]. Activities on other NMDA receptor subtypes, i.e., GluN1, GluN2A, 2C, and 2D, indicate poor selectivity for GluN2B which may cause numerous adverse effects. Even though there has been no success in the development of GluN2B-NMDA receptor antagonists thus far, such efforts are on the rise due to possibilities for treatment of various CNS diseases and deepening research on GluN2B-NMDA receptors [111]. Earlier research also discussed the capacity of NMDAR antagonists to influence synaptic plasticity; this was shown in treatment with RL-208, which significantly changed NMDAR2B and resulted in an elevation of various synaptic plasticity markers, including SYN, PSD-95, and Tgf [112].

NVP-AAM077 (now known as NVP) (Figure 4) was the first known selective GluN2A antagonist, but it had poorer selectivity for GluN2A over GluN2B in rodents [113] than had been obtained from recombinant human proteins that showed a 10-fold preference for GluN2A over GluN2B. This poor selectivity causes difficulty in the selective blockade of GluN2A without interfering with the GluN2B blockade; therefore, results from NVP-AAM077 remain controversial [113]. The study of GluN2A is therefore more limited due to lack of antagonists selective for this subunit. However, in 2017, a series of compounds were designed based on ACEPC [114], which is a competitive antagonist for the NMDA receptor. Among these compounds, ST3 exhibited Ki values of 52 nM for GluN1/GluN2A receptors and 782 nM for GluN1/GluN2B receptors. This represents a significant improvement in comparison to the widely used GluN2A-selective antagonist NVP-AAM077, as ST3 demonstrates a 15-fold preference for GluN1/GluN2A receptors over GluN1/GluN2B receptors [115].

In 2010, TCN-201 (Figure 5), the most promising selective GluN2A was reported and showed no activity on GluN2B at 50 uM but displayed submicromolar potency at GluN2A [116]. Later in 2016, Volkmann et al. (2016) [117] synthesized MPX-004 (Figure 6) and MPX-007 (Figure 7) using the TCN-201 scaffold. They showed absence of inhibition on GluN2B or 2D at concentrations that showed complete inhibition of GluN2A activity. They had greater water solubility and potency compared to TCN-201. Nevertheless, their selectivity at a high concentration (10 uM) was lower than the one observed for TCN-201, since both MPX-004 and MPX-007 inhibited GluN2B current (at 10 uM) by 8% and 30%, respectively [113].

TCN-201, a GluN2A NMDAR antagonist, looks more promising as a superior pharmacological tool for investigating the roles of GluN2A, since it has higher selectivity for GluN2A compared to GluN2B [113,118], probably because of the interactions at the ligand binding domains (LBDs) of GluN1 and GluN2A subunits. TCN-201′s binding site location is proposed to be at the interface between the LBDs of GluN1 and GluN2A subunits [119]. Modulation of LBD interactions has shown promising therapeutic potential. These interactions can be utilized in the discovery of novel NMDAR-based therapeutics [120]. The main function of LBDs in NMDA receptor function has been shown in the past two decades by several studies. Muller et al. (2017) [121] also recently reported that the chemical compound 3-bromo derivative **5i** (*N*-{4-[(2-benzoylhydrazino) carbonyl] benzyl}-3 bromobenzene sulfonamide), a TCN-201 derivative, has 2.5-fold higher antagonist activities than TCN-201, thus showing the feasibility of improvements. In 2016, Hackos and colleagues [122] introduced a novel category of positive allosteric modulators, demonstrating remarkable specificity for GluN1/GluN2A receptors [122]. Through an extensive high-throughput screening process, they identified a set of highly selective compounds, including GNE-6901 and GNE-8324, from a pool of 1.4 million compounds. X-ray crystallography results indicated that these compounds bind to a site situated at the interface between the GluN1 and GluN2A subunit LBDs [80,123].

### 1.7. Alzheimer’s Disease

Alzheimer’s disease (AD) is a chronic neurodegenerative disease that exhibits cognitive function decline and memory loss and is associated with the aged population. The population of patients worldwide is projected to reach 152 million by 2050 [124]. Aging is the main risk factor for many neurodegenerative diseases/disorders. Cognitive capacity decline is a major feature of brain aging where neuronal numbers and functions decrease due to various factors. GluN2B subunit expression was shown to reduce with age which correlates with weakening and decline in cognitive function. Therefore, there is mounting evidence that the GluN2B subunit is an essential subunit that requires attention when considering dementias related to a vast majority of neurodegenerative diseases [125,126]. Moreover, expression of GluN2B subunit reduces with age in animal models, which corresponds with reduced and weak performance in cognition [127,128]; in the frontal cortex of aged humans, there is a decrease in NMDA receptors with pharmacological characteristics associated with the GluN1/GluN2B subtype compared to other subtypes [129]; GluN2B subunit overexpression in the forebrain of transgenic mice has been shown to have significant cognitive benefits [47]. Synaptic strength increase through induced LTP elicits a selective increase in phosphorylation of GluN2B [130,131]. These facts suggest that controlled GluN2B potentiation can provide a novel strategy for treatment of cognitive disabilities since NMDA receptor-expressing neurons are more vulnerable to insults associated with cognition [132]. Complementarily, GluN2B-selective antagonists have been found to be relatively well tolerated [133]. Traxoprodil (CP-101, 606) (Figure 8), an antagonist selective for GluN2B, demonstrated improvement of task performances in some animal tests, suggesting them for potential use as enhancers of cognition in AD [131,134]. The tests measured compulsive behavior and impulsive action, which could be the underlying emerging evidence that suggests that glutamate signaling via GluN2B NMDA receptor plays an essential role in behavioral flexibility [130]. Blockade by Ro-25-6981 (Figure 9) in the presence of excessive GluN2B-NMDAR expression has also shown enhancement in retrieving spatial memory in rats [135] and has facilitated formation of spatial memory in mice [107]. According to findings by Hanson and colleagues in 2020, the use of GluN2A-selective positive allosteric modulators such as GNE-0723 led to a reduction in abnormal low-frequency oscillations and epileptiform discharges, while also demonstrating an improvement in cognitive deficits in a mouse model of Alzheimer’s disease [88].

Although the pathophysiology of Alzheimer’s disease (AD) is not completely understood, the formation of extracellular plaques is widely recognized as a prominent characteristic. These plaques primarily consist of beta-amyloid peptides [136,137]. Under normal conditions of glutamate levels, the NMDA receptor plays a crucial role in memory acquisition and learning processes [138]. However, in AD, the beta-amyloid oligomers disrupt the uptake of glutamate and lead to an increase in its release. Consequently, this imbalance results in elevated levels of neurotransmitters within the synaptic cleft [139]. Chronic stimulation of NMDA receptors at moderate rates elicits excitotoxicity, loss of neurons, and the characteristic decline of memory and loss of cognitive abilities seen in AD patients [140]. Kakefuda et al. (2016) [141] reported a significant amelioration of deficits in the working memory of DGKb knockout mice in a Y-maze test when memantine was administered. The study also revealed that GluN2A and GluN2B subunit expression levels were increased in the pre-frontal cortex, even though there was a reduction in the hippocampal region of DGKb-knockout mice. These decreases in GluN2A and GluN2B expressions in the hippocampus were assumed to have contributed to spatial memory deficits and decreased LTP [141]. Memantine (Figure 10), however, brought about improvements in the working memory and reversed the altered expression levels of GluN2A and GluN2B subunits. There is contemplation that the GluN2B subunit is a potential target for treating neurodegenerative disorders such as AD, as it is regarded as the primary target of memantine. Suggested mechanisms of action of memantine show LTP improvement, increased levels of BDNF and its receptors, and GluN2B subunit upregulation [142]. Although it has brought improvement of cognitive and memory functions in patients, it has failed to undo the progression of the disease.

### 1.8. Parkinson’s Disease

Parkinson’s disease (PD) is characterized by the dysfunction and loss of neurons in the brain, particularly those in the substantia nigra that are responsible for dopamine production [143]. As the disease advances, there is a decline in dopamine production, leading to the degeneration of dopaminergic neurons in the neural structure known as the pars compacta [144]. There is an overlap of activities between dopaminergic neurons from substantia nigra pars compacta and glutamatergic neurons projecting into the striatum from the motor and sensory cortex on the GABAergic spiny projection neurons. It has been established that glutamatergic and dopaminergic signaling display interactions with one another in controlling motor functions; therefore, alteration of NMDARs is not a surprise in this disorder [145]. The glutamatergic pathways from the subthalamic nucleus to the basal ganglia output nuclei become overactivated and are thought to play key roles in the pathophysiology of the symptoms of PD because of the high density of GluN2B-containing NMDARs in the basal ganglia [146].

Several studies have demonstrated that NMDAR antagonists can improve PD symptoms, serve as neuroprotective agents, and delay PD progression by inhibition of excitotoxicity fueled by the glutamatergic system. GluN2B-selective antagonists like radiprodil (Figure 11), traxoprodil, and ifenprodil improve PD motor symptoms, increase antiparkinsonian efficacy of dopaminergic agents synergistically, and lower L-DOPA-induced dyskinesia [147], supporting the suggestion that the glutamatergic system may elicit anti-parkinsonian effects if properly manipulated. According to Wessel et al. (2004) [94], NMDA receptors containing the GluN2B subunit are believed to play a significant role in the negative changes observed in glutamatergic function following the loss of dopamine neurons. Their study demonstrated that the effects of CP-106,606 on rotational responses suggest that the activation of NMDA receptors containing GluN2B may contribute to the development and maintenance of reduced responsiveness to dopamine agonists with chronic therapy [148]. Therefore, NMDA receptor antagonists selective for GluN2B may offer potential therapeutic benefits for patients with Parkinson’s disease. They represent an appealing approach in the treatment of the primary symptoms and motor complications that show up at a later stage [94,149].

Some NMDA receptor antagonists selective for GluN2B subunit receptors have shown antiparkinsonian activity in MPTP-lesioned primates, either when used alone or in combination with levodopa [150], and reduce dyskinesias induced by levodopa [151,151]; CP-101,606, an NMDA receptor antagonist selective for GluN2B, has shown the ability to prevent the onset of motor response abnormalities induced by levodopa in Parkinsonian rats, as well as improve these motor deficits after they have developed. These findings are similar to the effects observed with the administration of non-selective NMDA receptor antagonists [152], indicating that the activation of NMDA receptors containing the GluN2B subunit alone can significantly contribute to the development and persistence of these motor response alterations. Recent evidence suggests that the GluN2B subunit plays a crucial role in the molecular mechanisms underlying Parkinson’s disease, opening up the possibility of GluN2B-selective antagonists as potential therapeutic options for treating neurodegenerative disorders [153].

### 1.9. Neuropathic Pain

Neuropathic pain arises from damage or malfunction of the nervous system, and its treatment has been challenging for physicians due to limited understanding of its underlying mechanisms [154,155]. In recent times, there has been a shift away from opioids as the primary choice for managing neuropathic pain. Instead, there is a growing focus on selective antagonists targeting pre-synaptic NMDA receptors, sparking significant interest in the field, which have been used intermittently for neuropathic pain [156]. Further investigations into these pharmacological agents are ongoing to evaluate their efficacy.

The involvement of NMDARs in neuropathic pain has been demonstrated through the use of non-selective and GluN2B-selective antagonists, which have shown effectiveness in alleviating neuropathic pain in animal models. Additionally, studies using knock-in mice with a mutation in GluN2A, resulting in reduced Zn^2+^ inhibition and increased GluN2A-NMDAR activity, have shown enhanced neuropathic pain, suggesting that such function may contribute to neuropathic pain. In-depth studies of pain have shown that central sensitization may play an essential role in developing and maintaining neuropathic pain [157]. NMDA receptor activation is necessary for triggering central sensitization in the spinal dorsal horn (DH), most especially the GluN2B subunit. The most common GluN2 subunit in the DH of the spinal cord in rats is GluN2B [158]. Additionally, GluN2A’s expression has been shown to drop selectively following nerve lesion, further augmenting GluN2B subunit’s influence [159]. Thus, NMDA receptors of the GluN2B subunit are key targets for neuropathic pain possibly being involved in transmission of pain and synaptic plasticity [160].

High-affinity GluN2B antagonists PD 174494 (Figure 12), PD 196860 (Figure 13), traxoprodil (CP-101, 606), and Ro 25-6981 have shown activities in several animal pain models [161] with a huge distinction between anti-hyperalgesia and side-effect doses [162]. Several traxoprodil derivatives which are potent GluN2B-selective antagonists were later discovered and have shown great potential for development [139]. More compounds that demonstrate selectivity for GluN2B subtypes and superior potency are being examined.

### 1.10. Depression

There is high susceptibility for suicide in depressed patients, partly due to complications emanating from stress. Spine loss in animals can be induced by stress and inflammation [163], which in turn trigger depression in humans. These observations have led some researchers to classify depression as a neurodegenerative disease [164]. Preclinically, GluN2B-selective antagonists have demonstrated antidepressant activity through the rescue of spines and reversing depression-like behavior.

Over the past three decades, the antidepressant-like effects of NMDAR antagonists have been documented [165]. Ongoing research indicates that these antagonists exhibit rapid antidepressant effects [166]. The physiological and pharmacological characteristics of NMDA receptors are influenced by the specific subunit composition, which plays a crucial role in the therapeutic mechanisms of different antidepressants [106]. There is growing evidence suggesting that the signaling systems involved in synaptic plasticity are responsible for the rapid antidepressant properties observed with NMDAR antagonists [167,168].

There is significant evidence which shows that selective antagonists of GluN2B subunit have essential roles in treating depression. A highly potent GluN2B-selective antagonist, Ro 25-6981, displayed antidepressant effects [169]. One trial of traxoprodil (CP-101, 606), another GluN2B-selective antagonist, was, however, stopped because of obvious dissociative side effects in some patients during infusion as well as cardiotoxicity [170], though it had exhibited anti-depressant effects in treatment-resistant depression (TRD) patients with a 60% level in response [171].

Compounds with selectivity towards specific subunits, such as NVP-AAM077 targeting the GluN2A subunit [172], and Ro256981 targeting the GluN2B subunit [173,174], have demonstrated antidepressant-like effects in mice and rats. Wang et al. (2012) [175] reported that elevated levels of GluN1, GluN2A, and GluN2B are expressed in the chick retina, specifically in the cerebral cortex, making them potential targets for pharmacological inhibition of spreading depression (SD). Notably, the GluN2A-selective NMDA receptor antagonist NVP-AAM077 exhibited impressive inhibition of SD, suggesting the significant involvement of GluN2A-containing receptors in the genesis and propagation of SD [175]. Current antidepressant medications often require prolonged treatment and are ineffective in approximately 30% of patients with major depressive disorder (MDD), necessitating the development of new, faster-acting, and longer-lasting antidepressants [166]. MK-0657 (CERC-301) (Figure 14), a selective GluN2B antagonist, at 4–8 mg/day orally administered to patients for 12 days for treatment-resistant depression (TRD) disclosed a remarkable recovery from depressive symptoms when compared with placebo, and without major unfavorable side effects [176]. Some other compounds like EVT-101 (Figure 15), an NMDAR GluN2B antagonist (NCT01128452), have also reached clinical trials, but results have not yet been made public [166]. 

Studies have shown that GluN2A levels in the amygdala of depressed patients are significantly elevated when compared with healthy controls, which suggest disruption of glutamate signaling at the NMDA receptors of the amygdala in the brain [177]. In addition, a reduction in the expression of GluN2A and GluN2B protein subunits has been observed in the post-mortem tissue of depressed patients at the anterior PFC [178]. Kabir et al. (2020) [47] recently reported reduced glutamate recognition sites after measuring the protein immunoreactivity of GluN1, GluN2A, GluN2B, and GluN2C subunits in the cerebellum and hippocampal tissues of post-mortem samples from 13 depressed patients. The results showed that the GluN2A levels were elevated significantly in the amygdala of the patients who were depressed compared to controls that were healthy, which suggests disruption of glutamate signaling at the NMDA receptors in the amygdala. GluN2A and GluN2B subunits both exist in the basolateral amygdala (BLA). Suggestions by pharmacological investigations have shown that amygdala-dependent learning needs stimulation of GluN1/GluN2B heterodimers [179] since the GluN2B subunit explicitly bestows a rich pharmacology with recognition sites that are distinct for exogenous allosteric ligands. In addition, receptors containing GluN2B compared favorably with other subtypes of NMDAR and seemed to preferentially contribute to pathological processes associated with glutamatergic pathways’ overexcitation, making them preferred targets [180]. Recently, esketamine, an enantiomer of ketamine (Figure 16) which works by targeting the NMDA receptor, was approved for TRD by the FDA. Treatments have shown that esketamine (Figure 17 and Figure 18) is a promising option for TRD with improvement in depression symptoms [181]. It represents the first new class of antidepressants to be FDA-approved in 3 decades.

## 2. Concluding Remarks

In conclusion, biochemical, structural, and functional studies have greatly enhanced our understanding of the therapeutic potential of NMDAR antagonists while also shedding light on their associated side effects. To be viable for therapeutic use, antagonists must exhibit selectivity, mitigating the risk of NMDAR over-stimulation, which can lead to excitotoxicity and neurodegeneration, while preserving normal receptor function to avoid intolerable side effects. Going forward, both new and previously established therapeutic agents with subunit selectivity should be considered for the treatment of neurodegenerative and related disorders. Notably, newly discovered subtype-selective antagonists that target GluN2A- and GluN2B-containing receptors are currently undergoing clinical trials, offering promising prospects for the treatment of various diseases. There remains a substantial demand for innovative pharmacological tools and molecular biological techniques to gain precise insights into the regulation and antagonism of GluN2A or GluN2B NMDAR subunits. These efforts hold the potential to enhance subunit-specific signaling, thus improving the therapeutic efficacy of this class of compounds in neurological diseases and beyond.

## Figures and Tables

**Figure 1 pharmaceuticals-16-01535-f001:**
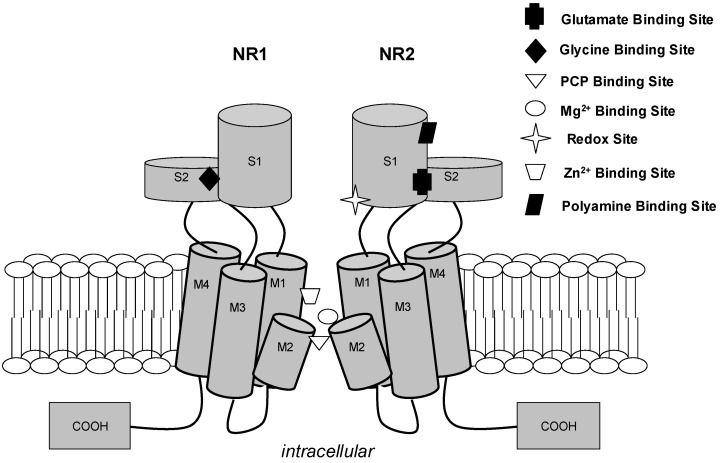
N-methyl-D-aspartate receptor cation channel. For clarity, two of four subunits (NR1 and NR2) of a functional channel are depicted. Each subunit contains three transmembrane regions (M1, M3, and M4) and a re-entry loop (M2), which is believed to form the channel selectivity filter influencing cation selectivity. The known agonists (glutamate and glycine), allosteric modulatory sites (Zn^2+^, polyamine, and redox sites), and uncompetitive antagonist sites (Mg^2+^ and PCP) are depicted at approximate proposed positions [11].

**Figure 2 pharmaceuticals-16-01535-f002:**
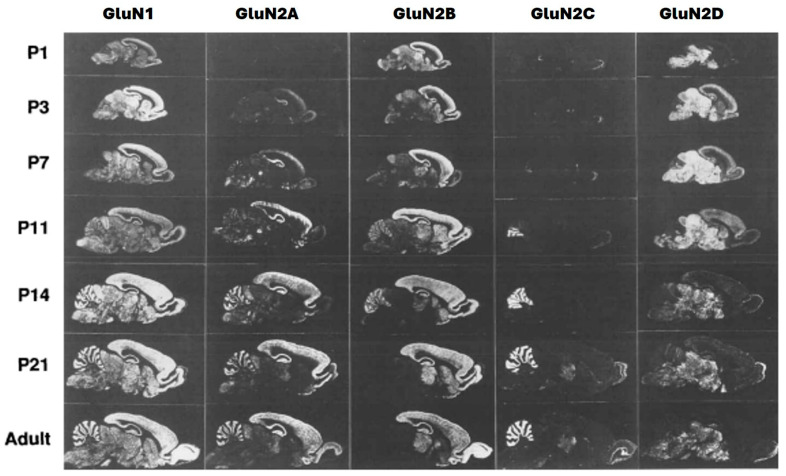
Expressions of GluN1 and GluN2A–D mRNAs in postnatal development (sagittal sections) [13].

**Figure 3 pharmaceuticals-16-01535-f003:**
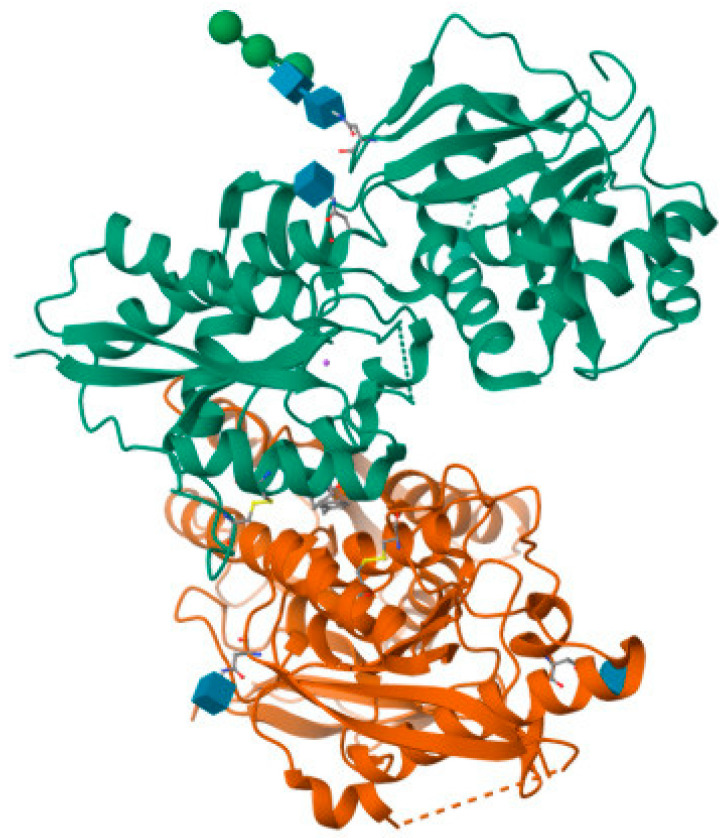
Crystal structure of amino terminal domains of the NMDA receptor subunits GluN1 and GluN2B in complex with ifenprodil (PDB ID: 5EWJ).

**Figure 4 pharmaceuticals-16-01535-f004:**
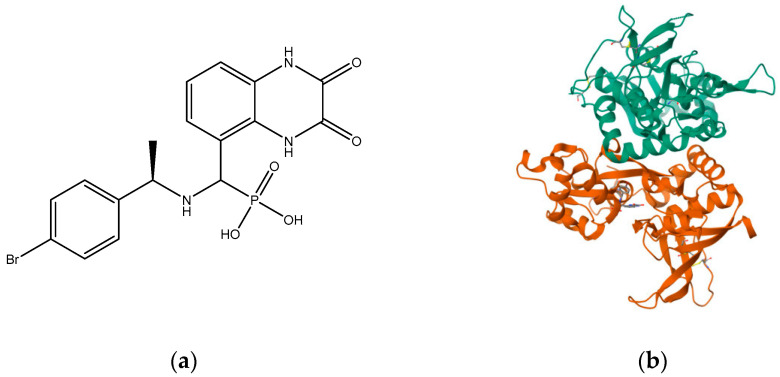
(**a**) NVP-AAM077; (**b**) crystal structure of GLUN1/GLUN2A ligand-binding domain in complex with glycine and NVP-AAM077 (PDB ID: 5U8C).

**Figure 5 pharmaceuticals-16-01535-f005:**
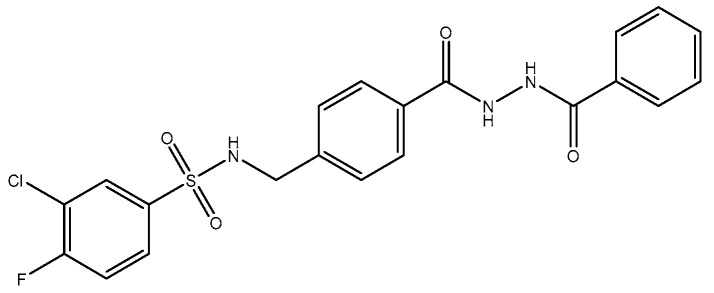
TCN.

**Figure 6 pharmaceuticals-16-01535-f006:**
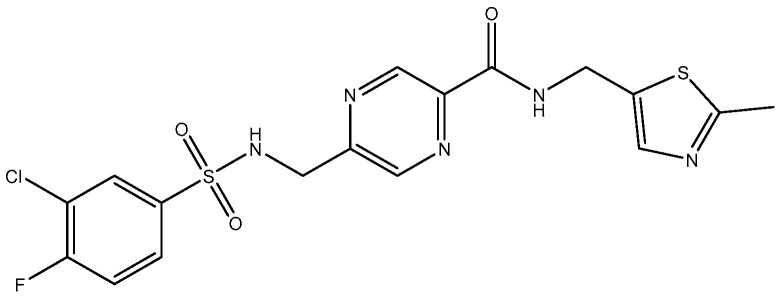
MPX-004.

**Figure 7 pharmaceuticals-16-01535-f007:**
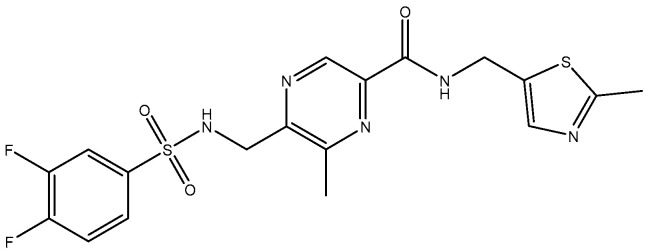
MPX-007.

**Figure 8 pharmaceuticals-16-01535-f008:**
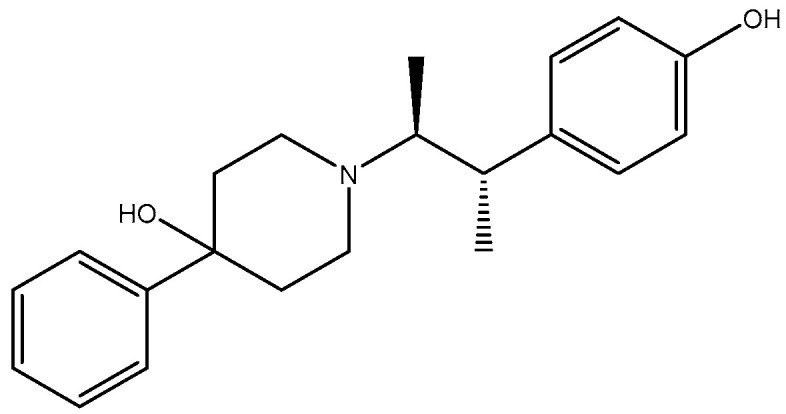
Traxoprodil.

**Figure 9 pharmaceuticals-16-01535-f009:**
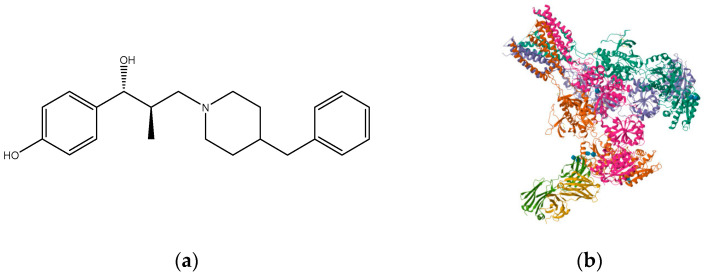
(**a**) Ro 25-6981; (**b**) triheteromeric NMDA receptor GluN1/GluN2A/GluN2B in complex with glycine, glutamate, Ro 25-6981, MK-801, and a GluN2B-specific Fab, at pH 6.5 (PDB ID: 5UP2).

**Figure 10 pharmaceuticals-16-01535-f010:**
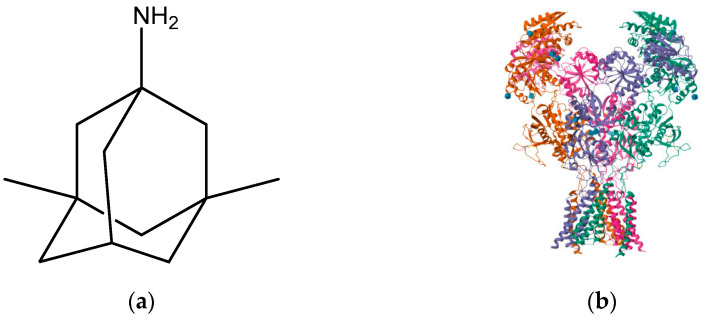
(**a**) Memantine; (**b**) memantine-bound GluN1a-GluN2B NMDA receptors (PDB ID: 7SAD).

**Figure 11 pharmaceuticals-16-01535-f011:**
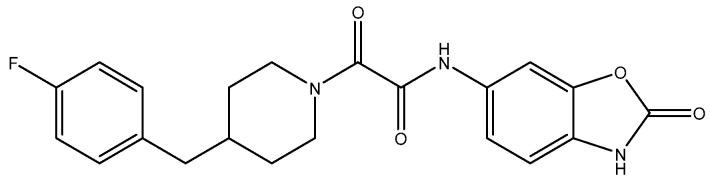
Radiprodil.

**Figure 12 pharmaceuticals-16-01535-f012:**
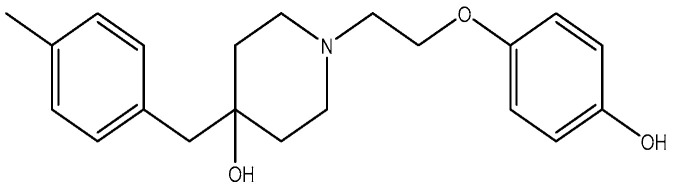
PD 174494.

**Figure 13 pharmaceuticals-16-01535-f013:**
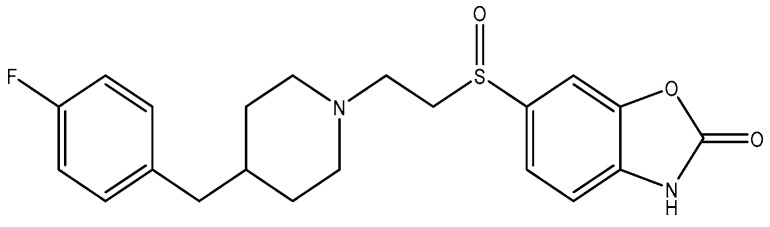
PD 196860.

**Figure 14 pharmaceuticals-16-01535-f014:**
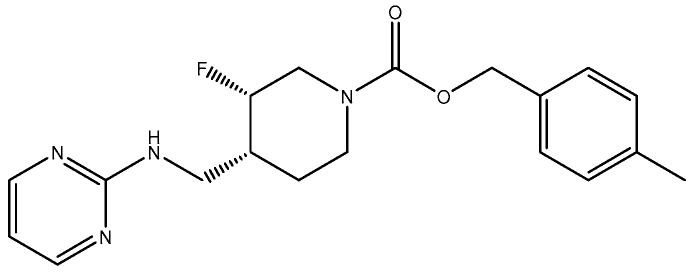
MK-0657 (CERC-301).

**Figure 15 pharmaceuticals-16-01535-f015:**
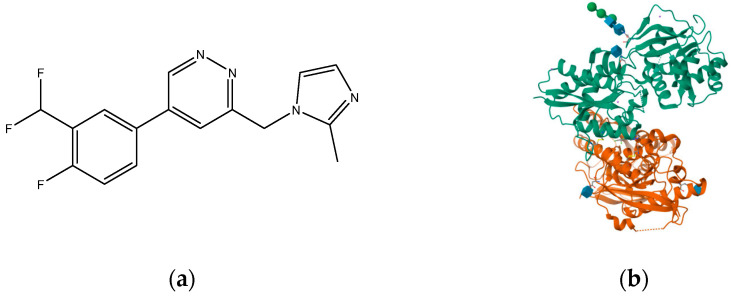
(**a**) EVT-101 (NCT01128452); (**b**) crystal structure of amino terminal domains of the NMDA receptor subunits GLUN1 and GLUN2B in complex with EVT-101 (PDB ID: 5EWM).

**Figure 16 pharmaceuticals-16-01535-f016:**
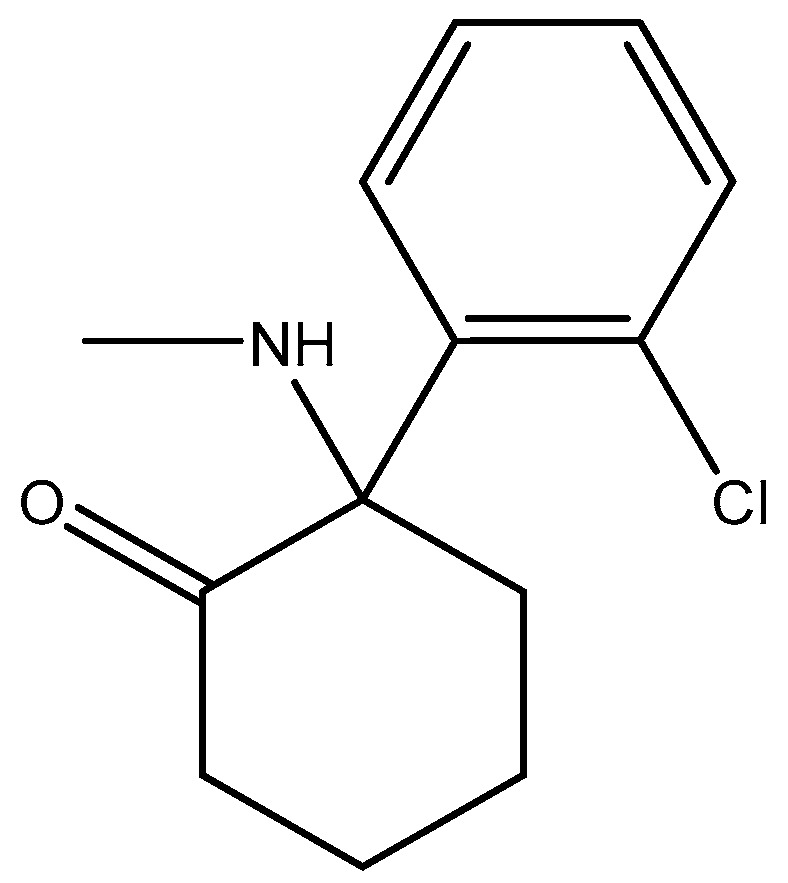
Ketamine.

**Figure 17 pharmaceuticals-16-01535-f017:**
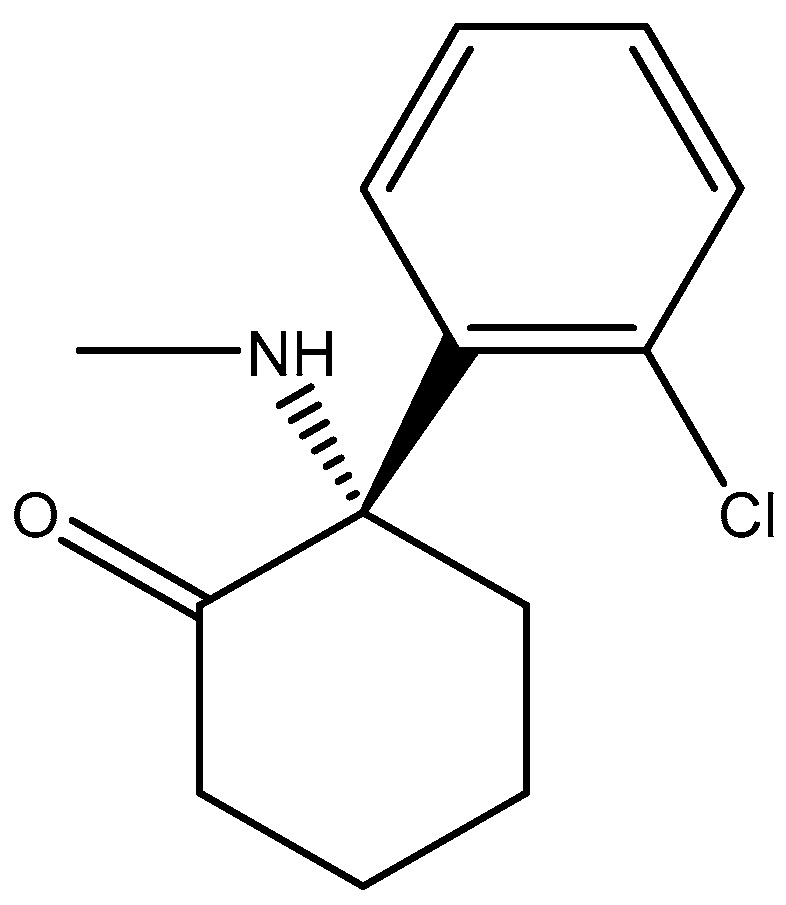
Esketamine.

**Figure 18 pharmaceuticals-16-01535-f018:**
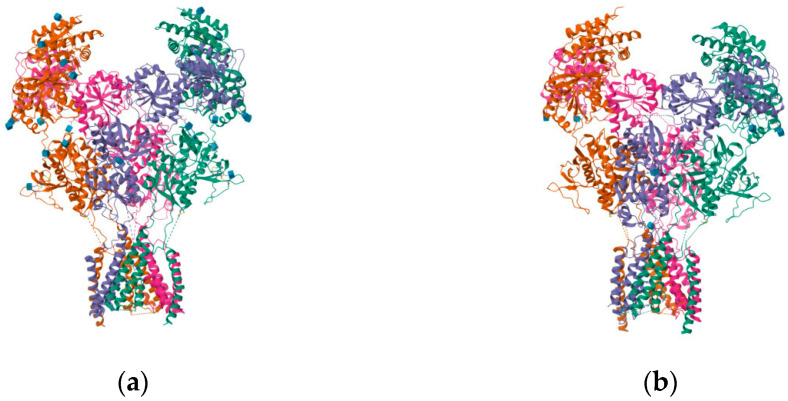
(**a**) Structure of the human GluN1-GluN2A NMDA receptor in complex with S-ketamine, glycine, and glutamate (PDB ID: 7EU7); (**b**) structure of the human GluN1-GluN2B NMDA receptor in complex with S-ketamine, glycine, and glutamate (PDB ID: 7EU8).

## Data Availability

Not applicable.

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
