# Peer review of "GluN2A and GluN2B N-Methyl-D-Aspartate Receptor (NMDARs) Subunits: Their Roles and Therapeutic Antagonists in Neurological Diseases"

_pharmaceuticals, 2023, doi:10.3390/ph16111535_

Round 1

Reviewer 1 Report

Comments and Suggestions for Authors

The review manuscript is organized primarily around NMDAR subunits and their roles in neurological and neuropsychiatric disorders.

Major comments

1.      P5: The section title 1.3. “GluN2A and GluN2B NMDA Receptor Antagonists in Neurodegenerative Diseases” is not appropriate.  This is because there are no or few descriptions about the association of NMDAR antagonists and neurodegenerative diseases. The only information related to NMDAR antagonists and neurodegenerative diseases is that ifenprodil (GluN2B inhibitor) could rescue spinal loss in HAND.  

2.      Based on the data described in the manuscript, the beneficial effects of GluN2B-containing NMDARs on cognitive function in aged subjects and the enhancement of hippocampus-dependent learning and memory by antagonizing GluN2B appear to be controversial. How do you explain this? This should be discussed more in the section.

3.      Throughout the manuscript, less information is available about GluN2A than GluN2B. As was the original purpose of this review, more information is also needed about GluN2A subunits and the application of their antagonists.

Minor comments

1.      P4: “…mixed populations of GluN1/2A, Glu1/2B and triheteromeric GluN1/2A/2B receptors.” “Glu1/2B” is missing N.

2.      P5: “Raybuck et al., (2017) (41) reported that Ifenprodil, reversed functional loss by rescuing spine loss in HIV-associated neurocognitive disorder (HAND)”. No comma between Ifenprodil and reversed.

Comments on the Quality of English Language

Moderate

Author Response

Reviewer 1:

Major comments

  1. P5: The section title 1.3. “GluN2A and GluN2B NMDA Receptor Antagonists in Neurodegenerative Diseases” is not appropriate.  This is because there are no or few descriptions about the association of NMDAR antagonists and neurodegenerative diseases. The only information related to NMDAR antagonists and neurodegenerative diseases is that ifenprodil (GluN2B inhibitor) could rescue spinal loss in HAND.  

Response to comment

The title has been corrected as suggested.

  1. Based on the data described in the manuscript, the beneficial effects of GluN2B-containing NMDARs on cognitive function in aged subjects and the enhancement of hippocampus-dependent learning and memory by antagonizing GluN2B appear to be controversial. How do you explain this? This should be discussed more in the section.

Response to comment

The data described needs detailed and extensive explanation of the experiment carried out in the study with drug dosages, hence controversial statement has been removed.

  1. Throughout the manuscript, less information is available about GluN2A than GluN2B. As was the original purpose of this review, more information is also needed about GluN2A subunits and the application of their antagonists.

Response to comment

More information about GluN2A has been added to the manuscript.

 Minor comments

  1. P4: “…mixed populations of GluN1/2A, Glu1/2B and triheteromeric GluN1/2A/2B receptors.” “Glu1/2B” is missing N.

Response to comment

Corrected as suggested

  1. P5: “Raybuck et al., (2017) (41) reported that Ifenprodil, reversed functional loss by rescuing spine loss in HIV-associated neurocognitive disorder (HAND)”. No comma between Ifenprodil and reversed.

Response to Comments

This has been corrected

Reviewer 2 Report

Comments and Suggestions for Authors Authors have summarized the roles of GluN2A and GluN2B N-Methyl-D-Aspartate receptor (NMDARs) subunits and therapeutic antagonists in Neurodegenerative Diseases. However, this review just summarized the role of NMDA receptors in neurodegenerative diseases. There are few references in the last 3 years and no recent progress on the function of NMDA receptors and their possible role in neurodegenerative diseases. Please add the  recent progress on the function of NMDA receptors and their possible role in neurodegenerative diseases and references in the last 3 years. 

In addition, the difference between extrasynaptic NMDAR and synaptic NMDAR and their possible role in neurodegenerative diseases should been reviewed.

Comments on the Quality of English Language

Authors have summarized the roles of GluN2A and GluN2B N-Methyl-D-Aspartate receptor (NMDARs) subunits and therapeutic antagonists in neurodegenerative diseases. However, this review just summarized the role of NMDA receptors in neurodegenerative diseases. There are few references in the last 3 years and no recent progress on the function of NMDA receptors and their possible role in neurodegenerative diseases.

In addition, the difference between extrasynaptic NMDAR and synaptic NMDAR and their possible role in neurodegenerative diseases should been reviewed. 

Author Response

Reviewer 2

  1. Authors have summarized the roles of GluN2A and GluN2B N-Methyl-D-Aspartate receptor (NMDARs) subunits and therapeutic antagonists in Neurodegenerative Diseases. However, this review just summarized the role of NMDA receptors in neurodegenerative diseases. There are few references in the last 3 years and no recent progress on the function of NMDA receptors and their possible role in neurodegenerative diseases. Please add the recent progress on the function of NMDA receptors and their possible role in neurodegenerative diseases and references in the last 3 years. 

Response to Comment

This has been worked upon, recent progress on the function of NMDA receptors and their possible role in neurodegenerative diseases and references in the last 3 years have been added. 

  1. In addition, the difference between extrasynaptic NMDAR and synaptic NMDAR and their possible role in neurodegenerative diseases should been reviewed.

Response to Comment

Extrasynaptic NMDAR and synaptic NMDAR and their possible role in neurodegenerative diseases have been reviewed and added to the manuscript

Reviewer 3 Report

Comments and Suggestions for Authors

The review is interesting and originality is conferred by the choice of compounds specifically acting as antagonist in NMDA subunits. 

-Check the entire manuscript to aovid typing, grammar and punctuation mistakes. 

- Due to reported compounds in protein data bank on NMDA-NxA subUnits, your manuscript could be enriched by data and figures obtained from these reports. No expert edition is required, just take a look in the PDB for these human structures. 

- Please check on the very recent reports. There are some relevant ligands recently reported as antagonist/blockers in those subunits. All linked to neurodegenerative processes.

- Conclusions should be clear regarding the scope and potential applications in neurological diseases or in the biomedical field.

Comments on the Quality of English Language

Punctuation and grammar mistakes.

Author Response

Reviewer 3

  1. Check the entire manuscript to avoid typing, grammar and punctuation mistakes. 

Response to Comment

Manuscript has been checked for mistakes and corrected.

  1. Due to reported compounds in protein data bank on NMDA-NxA subUnits, your manuscript could be enriched by data and figures obtained from these reports. No expert edition is required, just take a look in the PDB for these human structures. 

Response to Comment

Data from PDB has been added to the manuscript

  1. Please check on the very recent reports. There are some relevant ligands recently reported as antagonist/blockers in those subunits. All linked to neurodegenerative processes.

Response to Comment

Recently reported antagonist/blockers in the subunits have been added.

  1. Conclusions should be clear regarding the scope and potential applications in neurological diseases or in the biomedical field.

Response to Comment

This has been addressed.

Further additions and corrections have been highlighted in the manuscript, and I am willing to provide further clarification or address any additional questions from the editor or referees.

Round 2

Reviewer 1 Report

Comments and Suggestions for Authors

The manuscript has significant improvement in both content and quality.

My suggestion is to change the Title to "GluN2A and GluN2B N-methyl-D-arpartate Receptor (NMDAR) Subunits: Their roles and Therapeutic Antagonists in Neurological diseases".  

Comments on the Quality of English Language

N/A

Author Response

Comments and Suggestions for Authors: The manuscript has significant improvement in both content and quality.

My suggestion is to change the Title to "GluN2A and GluN2B N-methyl-D-arpartate Receptor (NMDAR) Subunits: Their roles and Therapeutic Antagonists in Neurological diseases".  

RESPONSE TO REVIEWER 1

The title has been changed and highlighted in the text.

Reviewer 2 Report

Comments and Suggestions for Authors  

Authors have revised the manuscript and added the recent progress on the function of NMDA receptors and their possible role in neurodegenerative diseases. Adding the recent progress to each section instead of presenting them as a separate part might be better. 

Comments on the Quality of English Language

No comments.

Author Response

Comments and Suggestions for Authors: Authors have revised the manuscript and added the recent progress on the function of NMDA receptors and their possible role in neurodegenerative diseases. Adding the recent progress to each section instead of presenting them as a separate part might be better. 

RESPONSE TO REVIEWER 2

Recent progress has been added to each section as suggested and highlighted in the text.

Reviewer 3 Report

Comments and Suggestions for Authors

Authors have addressed my suggestions and comments. Manuscript could be considered for publishing.

Comments on the Quality of English Language

Minor grammar details.

Author Response

Comments and Suggestions for Authors: Authors have addressed my suggestions and comments. Manuscript could be considered for publishing.

Minor grammar details

RESPONSE TO REVIEWER 3

Minor grammatical details noted and corrected.